# Enhanced Removal of Bordeaux B and Red G Dyes Used in Alpaca Wool Dying from Water Using Iron-Modified Activated Carbon

Gilberto J. Colina Andrade *[ID], Jessica M. Vilca Quilla, Ruly Terán Hilares[ID], Kevin Tejada Meza[ID], Alejandra C. Mogrovejo Valdivia, Jorge A. Aguilar-Pineda[ID], Jaime D. Cárdenas García and David A. Pacheco Tanaka

Laboratorio de Tecnología de Membranas, Universidad Católica de Santa María—UCSM, Urb. San José, San José S/N, Yanahuara, Arequipa 04000, Peru; jvilcaquilla@gmail.com (J.M.V.Q.); rteran@ucsm.edu.pe (R.T.H.); ktejada@ucsm.edu.pe (K.T.M.); amogrovejov@ucsm.edu.pe (A.C.M.V.); jaguilar@ucsm.edu.pe (J.A.A.-P.); jcardenas@ucsm.edu.pe (J.D.C.G.); alfredo.pacheco@ucsm.edu.pe (D.A.P.T.)
* Correspondence: gcolina@ucsm.edu.pe; Tel.: +51-918783027

**Abstract:** The aim of this research was to explore the removal of Red G and Bordeaux B dyes from water using a packed bed column with conventional carbon ($C_{-conv}$) and iron-modified activated carbon ($C_{-FeCl_3}$). The bands increased in $C_{-FeCl_3}$, corresponding to groups already existing in $C_{-conv}$, such as C = C and C-C, and the appearance of new groups, such as C-O, C-Cl, Fe-Cl and Fe-O. The total ash content ($C_T$) was $C_T$ = (10.53 ± 0.12 and 8.98 ± 0.21)% for $C_{-conv}$ and $C_{-FeCl_3}$, respectively. A molecular structure in the shape of a cross was noticed in Bordeaux B, which was less complex and smaller than the one in Red G. For fixed-bed columns, the carbon fraction was (0.43 and 0.85) mm. The pH of the adsorbents was 8.55 for $C_{-conv}$ and 4.14 for $C_{-FeCl_3}$. Breakthrough curves were obtained and the Thomas model (TM) and Yoon–Nelson model (YNM) were applied. The sorption capacity of Bordeaux B on $C_{-conv}$ and $C_{-FeCl_3}$ was $q_{TH}$: (237.88 and 216.21) mg/g, respectively, but the one of Red G was $q_{TH}$: (338.46 and 329.42) mg/g. The dye removal ($R_T$) was over 55%.

**Keywords:** iron-modified activated carbon; azo dyes; fixed-bed columns; adsorption



## 1. Introduction

The textile industry uses the largest amount of dye, accounting for approximately 700,000 tons per year worldwide [1]. Moreover, only 47% of organic dyes in wastewater is biodegradable because they have a complex molecular structure [2]. Therefore, the wastewater generated contains dyes and effluent discharge, which cause an environmental problem requiring new processes to remove efficiently. Common methods reported correspond to the adsorption process [3], membranes [1], AOP [4] and biological processes using microalgae, bacteria and fungi [5].

The adsorption process consists of a mass transfer of pollutants from a fluid to an adsorbent surface [6]. The advantages of the adsorption process over other technologies for the removal of pollutants include a low cost, high effectiveness without producing secondary waste and ease of design and operation [7]. Activated carbon has been shown to be an excellent adsorbent in various studies due to its high porosity, great affinity with many types of pollutants and large specific surface area [8]. Usually, however, the intrinsic characteristics of pure activated carbon are not adequate enough to remove pollutants favorably.

Therefore, the activated carbon must be subjected to modification processes. The modification of activated carbon is performed for various purposes, including promoting its separation properties through impregnation with magnetic particles and enhancing its sorption capacity by incorporating nanomaterials [9,10].

Iron has been used to modify different adsorbents, including biochars, zeolites, montmorillonite and activated carbon [9]. When activated carbon is added to a precursor solution containing iron ions, the product is an iron-activated carbon composite. The ions can diffuse deeply into the internal pores of activated carbon and link with its surface functional groups, combining the advantages of both materials. Using iron-oxide-decorated activated carbon as an adsorbent may allow for the user to benefit from the advantages of both materials [11].

Azo dyes are extensively used in the textile industry, and their release into the environment is a severe problem. In this study, activated carbon with added iron chloride was used to remove dyes such as Bordeaux B and Red G using fixed-bed columns. Mathematical models of BDST, Yoon–Nelson and Thomas were applied.

## 2. Methods

### 2.1. Reagents

The LANASET Bordeaux B and Red G dyes were supplied by a local textile company in Arequipa Perú. FILTRASORB 200 granular activated carbon and iron chloride hexahydrate ($FeCl_3.6H_2O$) were obtained for analysis from Emsure®.

The optimized molecular structures and functional groups of the dyes studied are presented in one- and three-dimensional (3D) planes in Figures 1–3 (GaussView 6 Program; Gaussian 16 Program, Wallingford, CT, USA). The visualization and quantum calculation of the dimensioning of these structures allowed for the establishment of differences regarding their shapes and sizes (VMD and Gaussian 19 programs, respectively) [12]. Considering the degree of complexity according to their structure (from highest to lowest) and the calculation of their molecular weights, the following order was established: Bordeaux B (1082 g/mol) > Red G (832 g/mol). Depending on their composition, these molecules were azo dyes with very complex conjugation structures $\pi$ ($N = N$) naphthalenes and sulfonic groups $SO_3^-$, interacting noncovalently with atoms such as $Na^+$ and $Cr^{+3}$. These functional characteristics presented some similarities with less complex conventional dye structures used in other studies. These molecules denoted resistance to natural biodegradation due to the aromatic rings present in their structure.

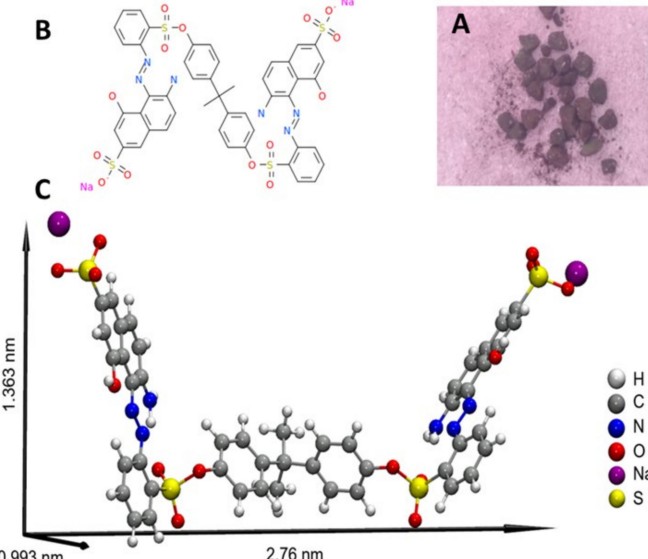

**Figure 1.** Molecular structure of the Bordeaux dye B: (**A**) photo of the dye shown in granular form, (**B**) image of the one-dimensional molecular structure and (**C**) image of the molecular structure dimensioned in 3D.

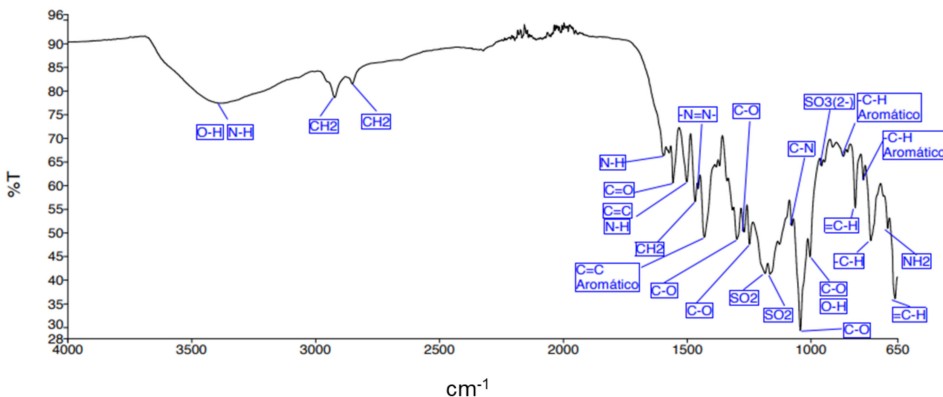

**Figure 2.** Infrared spectrum of the Bordeaux B dye identifying the bands with their functional groups.

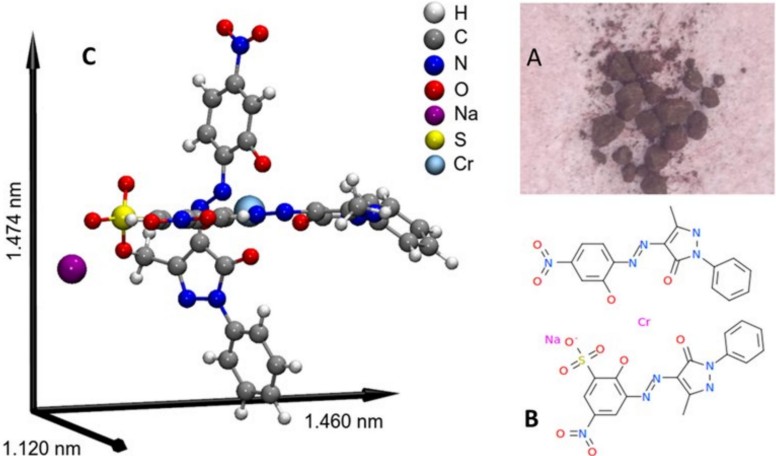

**Figure 3.** Molecular structure of the Red G dye: (**A**) photo of the dye shown in granular form, (**B**) image of the one-dimensional molecular structure and (**C**) image of the molecular structure dimensioned in 3D.

The structure and molecular size are important factors that must be considered when implementing degradation and/or removal treatments using adsorptive processes; in this way, the global speed of molecular migration from the fluid where it is dissolved can be inferred up to the surface of the adsorbents [13].

The chemical composition and nonbiodegradable nature of some of the dyes used in the industry can generate derivatives with a certain degree of toxicity, carcinogens, teratogens or mutagens. Their resistance to oxidation and chemical stability give them a sustained persistence in the environment and make them highly toxic organic pollutants. Furthermore, these pollutants are difficult to remove using conventional wastewater treatment technologies [14]. Also, Figure 3 shows an image of the Bordeaux B dye, photographed in its granular form (A), one-dimensional molecular structure (B) and three-dimensional molecular structure (C).

A complex molecular structure was noticed in the shape of an inverted "M", with height, width and depth dimensions of 1363, 2.76 and 0.993 nm, respectively, with the presence of azo groups $\pi$ (N = N), naphthalenes and sulfonic groups $SO_3^-$, interacting in a noncovalent way, in this particular case with atoms such as $Na^+$. Figure 2 shows the infrared spectrum of this dye.

The qualitative analysis of infrared spectroscopy carried out on the Bordeaux B dye confirmed the identification of these functional groups among others present in their molecular structure, and the interaction of these groups in adsorptive processes was inferred. Considering the wave number $(1/\lambda)$, among the functional groups present were

C-O (1/$\lambda$ = 1298.64 1269.39 y 1247.43) cm$^{-1}$, SO$_3$ (1/$\lambda$ = 956.07 cm$^{-1}$), C=C Aromatic (1/$\lambda$ = 1429.57 cm$^{-1}$) and C=O (1/$\lambda$ = 1556.47 cm$^{-1}$) y SO$_2$ 1/$\lambda$ = (1184.15 y 1164.81) cm$^{-1}$ y -N=N- a 1/$\lambda$ = 1454.90 cm$^{-1}$.

Figure 3 shows the image of the Red G dye, photographed in its granular form (A), one-dimensional molecular structure (B) and three-dimensional molecular structure (C).

A molecular structure in the shape of a cross was noticed, a little less complex and smaller with respect to the previous dye, with height, width and depth dimensions of 1474, 1460 and 1120 nm, respectively, similarly with the presence of azo groups $\pi$ (N = N), naphthalenes and sulfonic groups $SO_3^-$, interacting in a noncovalent way, in this particular case with atoms such as $Na^+$. Figure 4 shows the infrared spectrum of this dye.

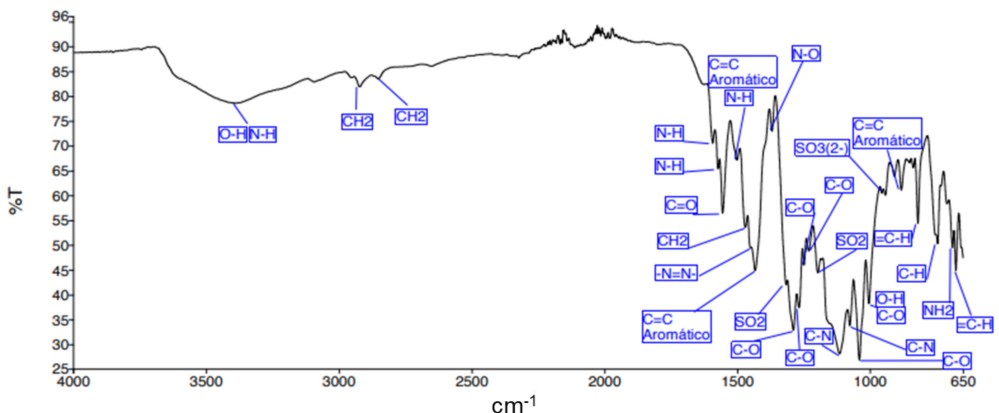

**Figure 4.** Infrared spectrum of the Red G dye identifying the bands with their functional groups.

The qualitative analysis of infrared spectroscopy carried out on the Red G dye confirmed the identification of these functional groups. Among others present in their molecular structure, the interaction of these groups in adsorptive processes was inferred. Considering the wave number (1/$\lambda$), the functional groups were similar to those contained in Bordeaux B dye, namely, C–O (1/$\lambda$ = 1298.64 1269.39 y 1247.43) cm$^{-1}$, SO$_3$ (1/$\lambda$ = 956.07 cm$^{-1}$), C=C Aromatic (1/$\lambda$ = 1429.57 cm$^{-1}$) and C=O (1/$\lambda$ = 1556.47 cm$^{-1}$) y SO$_2$ 1/$\lambda$ = (1184.15 y 1164.81) cm$^{-1}$ y -N=N- a 1/$\lambda$ = 1454.90 cm$^{-1}$.

Castillo-Cervantes et al. [15] found characteristic bands of secondary amines at 1/$\lambda$ = 3453, 1589, 1225 and 625 cm$^{-1}$, when they worked with the RO16 dye. Likewise, bands corresponding to aromatic rings and aliphatic chains (CeH stretch) at 1/$\lambda$ = 3080, 2986, 28.73 and 1053 cm$^{-1}$ and other very important bands appeared for the carbonyl group at 1/$\lambda$ = 1677 cm$^{-1}$, for the sulfonic acid group at 1/$\lambda$ = 1136 cm$^{-1}$ and, finally, the hydroxyl group at 1/$\lambda$ = 1020 cm$^{-1}$. Many of these functional groups coincided with the structures of the dyes addressed in this study. The bands observed in the region from 1650 to 1850 cm$^{-1}$ (approximately 1621 cm$^{-1}$) corresponded to a stretch of the C=O group, and the bands at approximately 1384 cm$^{-1}$ indicated the presence of alkenes [13,16].

### 2.2. Methods

### 2.2.1. Quantum Calculation Details

To obtain the sizes of dye molecules, DFT calculations were carried out to obtain the optimized structures. Initial structures were built using the Gauss View v.6 software package, USA and the quantum calculations were performed with the Gaussian 16 software package at the CAM-B3LYP/TZVP level of theory. The vibrational frequencies were calculated to ensure that the geometries were those of minimum energy. We used the SCRF method (SCRF = (SMD, solvent = water)) to include the water solvent effect. The box simulation dimensions were calculated using our own computational routine that performed a search for the most distant atoms. The reported measurements corresponded to the simulation box dimensions containing all the particles considering only their nuclear positions.

2.2.2. Treatment of the Dye Solution in a Continuous-Flow Column with Activated Carbon

Activated carbon was washed and dried in an oven at 80 °C for 24 h; later, its apparent density was determined in the front called $C_{-conv}$; then, 15 g of $C_{-conv}$ was transferred to a container containing 200 mL of $FeCl_3.6H_2O$. Subsequently, it was stirred at 250 rpm for approximately 24 h, and then it was filtered in a vacuum filtration system using a Whatman filter qualitative, after which the filtering was discarded. The modified carbon $C_{-FeCl_3}$ was placed in an oven at 80 °C for 24 h, and then stored in a desiccator until its use [17].

The determination of the particle size distribution (PSD) and the effective average particle diameter (EAPD) was carried out by means of the standard procedure (ASTM, 2010). A quantity of 100 g of FILTRASORB 200 granular activated carbon was weighed; then, it was placed on the upper sieve of a set of 8 × 30 stacked sieves with a mesh opening of 1.70, 1.18, 0.85, 0.425 and 0.250 mm, increasing the opening of the sieve from top to bottom, where it was stirred for 10 min. After time elapsed, the stacked sieve assembly was removed and the mass of the fraction retained on each sieve was recorded separately. It was verified that the sum of the masses of all the fractions did not deviate by more than 2.0 g from the initial mass value of the solid material. The fraction retained on each sieve was determined according to Equation (1):

$$R = (F/S) \times 100 \tag{1}$$

where R is the percentage retained in each fraction, F is the mass fraction retained on a sieve (g) and S is the sum of all the mass fractions retained on each sieve (g). The effective average particle diameter was calculated according to Equation (2):

$$EAPD_{effective} = \sum_{i=1}^{n} \left( \frac{F_i}{S} \times N_i \right) \tag{2}$$

where $EAPD_{effective}$ is the effective average particle diameter (mm), N is the factor for a given sieve (mm) and n is the number of sieves. The range of particle sizes used for each solid material in the execution of the tests was from 0.45 to 0.83 mm.

Apparent Density and Ash Content of Activated Carbon on Dry Basis

The determination of the apparent density and ash content total on a dry basis was carried out using the standard procedure [18].

pH Activated Carbon

The determination of the pH was carried out using the standard procedure [19].

Preparation of Activated Carbon and Packing of the Column

The treatments were carried out using two types of activated carbon for each dye under study. One of them was a commercial carbon, which, from now on, is referred to as $C_{-conv}$, and the other one was the same commercial carbon modified with $FeCl_3$, which, from now on, is referred to as $C_{-FeCl_3}$.

The sieved activated carbon, the one whose particle diameter (*pd*) range was between 0.60 and 0.85 mm, was taken to ensure a ratio ($\phi_{col}/pd$) greater than 12 in order to avoid the wall effect that could affect the breakthrough performance [20,21]. The charcoal was washed thoroughly with deionized water until all the dust was removed and the pH of the wash water did not show significant changes. Then, the charcoal was dried in the oven at 150 °C for 4 h and stored in an amber.

Column Packing

For the packing of the column, the concentric column of a distillation tube was selected, whose dimensions were approximately 30 cm in height and with a 1.10 cm internal diameter ($\phi_{col}$) composed of borosilicate glass. Before being packed with the selected activated carbon ($C_{-Conv}/C_{-FeCl_3}$), a small amount of glass wool was placed at its lower end, which served to

support the activated carbon bed and also to provide a flat surface across the diameter of the column. Similarly, a small amount of glass wool was placed on the top of the column once packed.

With an accuracy of 0.1 mg, approximately 5 g of deionized water was added to the activated carbon and placed in a beaker with constant stirring and allowed to heat to a boil for 10 min in order to replace the air in the pores of the activated carbon with water. After cooling to room temperature, the airless carbon was transferred to the column in such a way as to exclude air bubbles from the column, until a bed height of approximately 10 cm was obtained.

A 2 L container served as the system feeder. Its content was a solution of the dye (the solutions of Bordeaux B and Red G were prepared separately from the solid powder (Merck, Darmstadt, Germany)). The initial nominal concentration of each dye was predetermined at 50 mg/L (adsorbates). The solutions had an initial pH of 6.39 and 6.96 for Bordeaux B and Red G, respectively. Each dye solution was separately passed through the column with an flow rate of 1.30 mL/min by means of a peristaltic pump (Masterflex, Radnor, PA, USA). The process temperature was $25 \pm 0.1\ °C$, adjusted by the recirculation of water provided by the thermostatic bath.

Packed-Bed Column Operation

For the operation of the column, once it started, samples of the column effluent were periodically collected in tubes (13 mL capacity), and their concentrations were analyzed with a UV–visible spectrophotometer mark GENESYS 150 at $\lambda$, determined using quartz cells, to obtain the concentrations of each colorant in the effluent or residual colorant concentration. The sampling frequency was carried out every 2.5 mL of effluent in the first eight intakes, and from then on, they were taken every 5 mL.

The operation of the column was carried out until the concentration of the dye at the exit of the treatment was approximately equal to the entrance concentration (Figure 5). Breakthrough curves were developed for each dye, plotting the effluent concentration ($C_{ef}$) in mg/L or the relationship ($C_{ef}/C_0$) versus effluent volume ($V_{ef}$) in mL or time (t) in min.

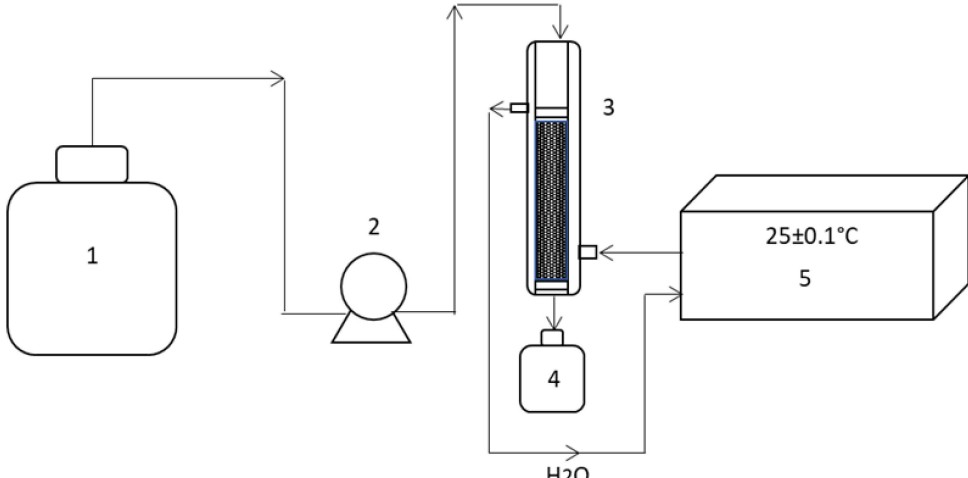

**Figure 5.** Experimental setup of the continuous-flow adsorption system. (1) Feeder container. (2) Peristaltic pump. (3) Column packed with granular activated carbon $C_{-conv}/C_{-FeCl_3}$. (4) Collection container. (5) Water recirculation bath at $T = 25 \pm 0.1\ °C$.

The removal percentage was calculated with Equation (3):

$$E = \frac{(Co - Ce)}{Co} * 100 \qquad (3)$$

where Co is the initial dye concentration in the solution (mg/L) and Ce is the final concentration (mg/L) [10].

Estimation of the Adsorption Column Design Parameters

Certain parameters are fundamental to establish the adsorption dynamics in a packed bed for large-scale applications, such as the adsorption capacity at breakthrough and saturation [22,23]. The effluent volume $V_{ef}$ (mL) was calculated using Equation (4):

$$V_{ef} = Qt_{total} \tag{4}$$

where $t_{total}$ and $Q$ are the total flow time (min) and the volumetric flow (mL/min), respectively. The area under the breakthrough curve (A) obtained by the integration of the adsorbed $C_{ad}$ adsorbate concentration (mg/L) versus $t$ (min) of the breakthrough curve could be used to calculate the total amount of adsorbate adsorbed on the column $q_{total}$ (mg), for some operating parameters determined according to Equation (5):

$$q_{total} = \frac{QA}{1000} = \frac{Q}{1000} \int_{t=0}^{t=t_{total}} C_{ad}dt \tag{5}$$

The total amount of adsorbate sent to the $m_{total}$ column (mg) was calculated using Equation (6):

$$m_{total} = \frac{C_0 Q t_{total}}{1000} \tag{6}$$

The percentage of the total removed $R_T$ (column performance) was calculated with Equation (7):

$$R_T = \frac{q_{total}}{m_{total}} \times 100 \tag{7}$$

The length of the mass transfer zone ($L_{MTZ}$, cm) was calculated using Equation (8):

$$L_{MTZ} = \left(\frac{ts - tb}{t_{total}}\right) * Z \tag{8}$$

where $ts$ and $tb$ are the saturation and breakthrough times (min), respectively, and $Z$ is the bed height (cm).

Kinetic Models

The adsorption column was subjected to axial dispersion, external film resistance and intraparticle diffusion resistance. The mathematical correlations for adsorption in fixed-bed columns were based on the assumption of axial dispersion, external mass transfer, intraparticle diffusion and nonlinear isotherms. Mathematical models were developed for the evaluation of the efficiency and applicability of the column models for large-scale operations. Two mathematical models were used for kinetic studies: the Thomas model (TM) and Yoon–Nelson model (YNM). The TM was proposed on the assumption of Langmuir kinetics of adsorption–desorption that rate driving forces following second-order reversible reaction kinetics and no axial dispersion. The YNM is a simple theoretical assumption, which does not concentrated on the properties of the adsorbate, type of adsorbent and any physical features of the adsorption bed. This model offers proof that the decreasing rate of adsorption is directly proportional to the adsorbate adsorption and the breakthrough on the adsorbent. The YNM assumes the symmetric nature of the breakthrough curve and neglects axial dispersion effects. The linear form of the model equations and the model parameters were presented in Equations (9)–(12) [6,22]. The linear column kinetic model equations were shown below [24,25]. Nonlinear and linear forms of the Thomas model were represented through Equations (9) and (10), respectively:

$$\frac{Ct}{Co} = \frac{1}{1 + e^{(\frac{K_{Th}}{Q}(q_0.m - Co.V))}} \tag{9}$$

$$ln\left[\left(\frac{Co}{Ct}\right) - 1\right] = \left(\frac{K_{Th}q_{o}m}{Q}\right) - \left(\frac{K_{Th}C_{o}V_{ef}}{Q}\right) \tag{10}$$

where:

$K_{Th}$ = Thomas rate constant, mL/mg.min;
$q_{o}$ = maximum capacity of adsorption, mg/g;
$V$ = volume of solution, mL.

The nonlinear and linear forms of the Yoon–Nelson model were represented using Equations (11) and (12), respectively:

$$\frac{Ct}{Co} = \frac{1}{1 - e^{K_{YN}(\tau - t)}} \tag{11}$$

$$ln\left(\frac{C_{t}}{C_{o} - C_{t}}\right) = K_{YN}(t - \tau) \tag{12}$$

where:

$K_{YN}$ = Yoon–Nelson rate constant, $\text{min}^{-1}$;
$\tau$ = time required for 50% adsorbate breakthrough or time when $Ct/Co = 0.5$, min.

## 3. Results and Discussion

### 3.1. Correlation Analysis of pH with Respect to the Concentration of the Azo Dyes

The correlation between the pH variable and the concentration of the dyes was analyzed (spectral scan $\lambda = 200$–1100 nm). The analysis of variance did not show significant differences ($p = 0.05$) with respect to the variation of absorbances and the pH values of the different dyes used.

### 3.2. Treatment in a Fixed-Bed Column with Continuous-Flow Activated Carbon

3.2.1. General Qualitative and Quantitative Physicochemical Characteristics of the Adsorbent

The commercially purchased adsorbent under study was activated carbon called *C-conv*, which was modified according to the protocol described above using FeCl₃ ($C_{-FeCl_{3}}$). The effective average particle diameter (EAPD, effective/mm), particle size distribution (PSD), apparent density ($\rho$/g/mL), total ash content (%), pH, infrared spectroscopy FTIR and X-ray fluorescence were determined. The particle size distribution was determined for the adsorbents $C_{-conv}$ and $C_{-FeCl_{3}}$, in both cases showing a very similar PSD.

Particles whose diameters ranged between 0.85 and 1.18 mm, equivalent to 80% of the total particles, were obtained. The distribution showed the formation of a slightly symmetrical wider wave or bell, which translated into little uniformities in terms of the particle diameter. For the study, in fixed-bed columns only the carbon fraction was taken, whose particle diameter was between (0.43 and 0.85) mm. The calculation of the EAPD$_{effective}$/mm was 1.1 mm.

The effect of modifying the particle size during the fixed-bed adsorption experiments had an important effect on the performance of the column. In the fine particle size range, the breakthrough curves followed a much more efficient profile than the larger particle size ranges, where the breakthrough time increased and the curves tended to lean towards the classic "*S*"-shaped profile.

Increasing the size of the adsorbent particles was shown to reduce the maximum bed capacity and breakthrough time. Small particles have a short diffusion path, allowing the adsorbate to penetrate deeper inside more quickly [26]. The time required to achieve saturation decreased with an increase in particle size. This was because the larger the particle size, the less active sites for sorption as a result of the reduced surface area of the adsorbent.

A small particle size represents a more available surface for adsorbate and adsorbent interactions. Consequently, it increased the adsorption capacity, the saturation time, the

removal efficiency and the volume of treated effluent. The $L_{ZTM}$ increased with the increase in particle size. As a result of the lower adsorption capacity at a larger particle size, more space would be required to achieve a high mass transfer [22].

The presence of inorganic components in the adsorbents could be determined by analyzing the content of total ash-$C_T$ [27]; these were $C_T$ = (10.53 ± 0.12 and 8.98 ± 0.21)% and cv = (1.14 and 2.28)% for $C_{-conv}$ and $C_{-FeCl_3}$, respectively, with statistically significant differences between the two related groups ($p$ = 0.025, $p < 0.05$) [27]. Higher content values of the inorganic material (ash content) were reported in other studies [13].

An average of $\rho$ = 0.51 ± 0.03 g/mL was obtained for $C_{-conv}$ and $\rho$ = 0.53 ± 0.01 g/mL for $C_{-FeCl_3}$ without statistically significant differences between the related groups ($p$ = 0.423, $p > 0.05$). These results coincided with those published by Radhika et al. [6]. The apparent density ($\rho$) provided information about the mechanical resistance of adsorbents when they were part of a fixed bed within an adsorption column. It was reported that values 0.3 g/mL or higher constituted materials with good mechanical resistance in continuous-flow processes [6].

The pH of the adsorbents was 8.55 ± 0.05 units for $C_{-conv}$ and 4.14 ± 0.03 units for $C_{-FeCl_3}$, with highly significant differences between the related groups ($p < 0.05$, $p$ = 0.00). It was reported that the pH of most commercial coals is due to inorganic constituents in the precursor, or those added during the manufacturing process. Activation conditions can also provide an explanation for the difference in pH observed between the coals. Basic pH values are characteristic of coals whose activation processes involve the use of $CO_2$. Such an explanation could be used for the case of $C_{-conv}$ used in this study, whose pH was approximately nine units [28].

Figure 6A,B shows the FTIR spectra of the activated carbons $C_{-conv}$ and $C_{-FeCl_3}$, respectively. The increase in bands in the modified carbon $C_{-FeCl_3}$ was evident, which corresponded to groups already existing in the conventional $C_{-conv}$, such as C = C and C–C, and the appearance of new groups, such as C–O, C–Cl, Fe–Cl and Fe–O, which made the $C_{-FeCl_3}$ modified carbon much more reactive on its surface. Typical FTIR spectrum of an activated carbon synthesized from a corn cob (raw corn cob), was published by Iheanacho et al. [22], where they obtained the formation of more visible bands of some compounds, such as carboxylic acids, ethers, esters and nitro compounds, after activation.

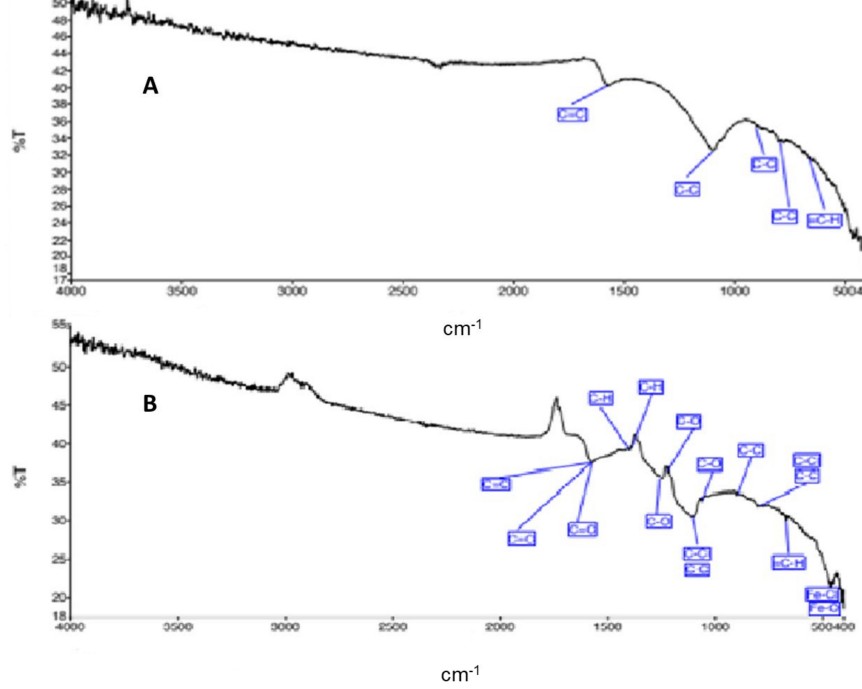

**Figure 6.** FTIR spectrum of activated carbons: (**A**) $C_{-conv}$ and (**B**) $C_{-FeCl_3}$.

The study of the FTIR spectra made it possible to clarify which functional groups were involved in the formation of intermolecular bonds [22]. The increase in the reactivity of the adsorption sites (Figure 6) and the surface charge at the edges of the adsorbates enhanced the electrostatic interaction between the two dyes and the adsorbents.

3.2.2. Treatment in a Fixed-Bed Column with Continuous-Flow Activated Carbon: Breakthrough Curve and Mathematic Models

Figure 7A,B shows the adsorptive performance of the Bordeaux B and Red G dyes during the treatment, describing the typical profile in the shape of an "*S*" (breakthrough curve) on the fixed bed packed with $C_{-conv}$ and $C_{-FeCl_3}$.

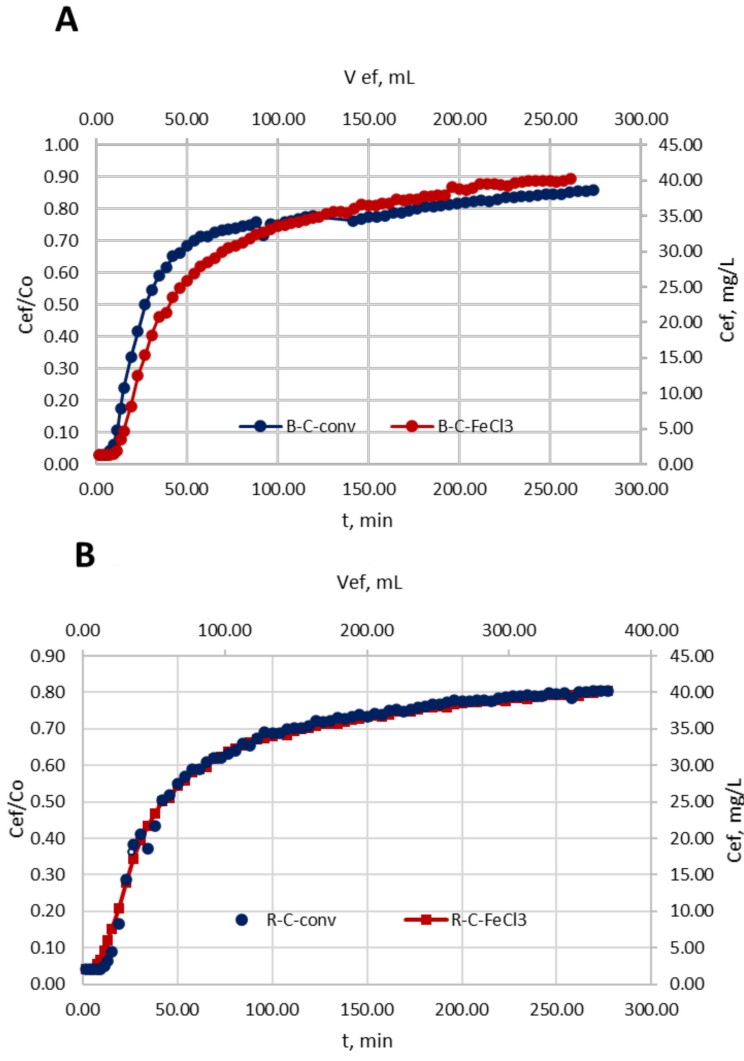

**Figure 7.** Observed breakthrough curve stains on $C_{-conv}$ and $C_{-FeCl_3}$ adsorbents. Average values are presented. (**A**) Bordeaux B and (**B**) Red G.

The average sum of errors (S) of all the experimental values along the breakthrough curve from $t = 0$, with $C_{-conv}$ and $C_{-FeCl_3}$, was $\Sigma \ \overline{X} \ S \ _{(Cef/Co)} = \pm 0.09$ and $\pm 0.02$, respectively, which showed little dispersion and reproducibility in the repetitions carried out.

The results of the evaluation parameters of the fixed-bed column packed with $C_{-conv}$ and $C_{-FeCl_3}$ during the adsorption process of Bordeaux B in the influent solution are shown in Table 1.

**Table 1.** Evaluation parameters of the fixed-bed column packed with $C_{-conv}$ and $C_{-FeCl_3}$ during the adsorption process of Bordeaux B in the influent solution.

| Adsorbent | $t_b$/min | $t_s$/min | $L_{ZTM}$/Cm | $q_{total}$/mg | $q_{eq}$/mg·g$^{-1}$ | $M_{total}$, mg | $R_T$/% |
|---|---|---|---|---|---|---|---|
| $C_{-conv}$ | 7.69 | 136.54 | 7.26 | 5.86 ± 1.29 | 1.17 ± 0.26 | 11.02 ± 5.19 | 56.78 ± 15.07 |
| $C_{-FeCl_3}$ | 10.56 | 198.05 | 8.43 | 7.93 ± 0.80 | 1.59 ± 0.16 | 13.98 ± 3.01 | 57.43 ± 6.66 |

Note: $t_b$, breakthrough time when $C_{ef} \approx 0.03C_0$. $t_s$, saturation time when $C_{ef} \approx 0.82C_0$.

The average values obtained from the *breakthrough* time ranged between 7.69 and 10.56 min for the removal on $C_{-conv}$ and $C_{-FeCl_3}$, respectively. The tb achieved in the removal of Bordeaux B on $C_{-FeCl_3}$ was approximately three minutes after that obtained with $C_{-conv}$. This implied that the classic "S" breakthrough profile started later for $C_{-FeCl_3}$, which translated into a longer life treatment for this bed—constant flow adsorbent. Regarding the averages of the saturation time (ts), a slight increase was observed from ts = 136.54 min reached with the $C_{-conv}$ ($V_{treated\ ef}$ = 177.50 mL) to ts = 198.05 min, obtained in the removal on $C_{-FeCl_3}$ ($V_{treated\ ef}$ = 257.50 mL).

The average lengths of the $L_{ZTM}$ mass transfer zone in the influent solution were found between 7.26 and 8.43 cm. However, small fluctuations in flow were probably the cause of the variations in the length values of the mass transfer zone. The adsorbed amount ($q_{total}$) was 7.93 ± 0.80 mg ($C_{-FeCl_3}$) higher than $C_{-conv}$. Similar values have been reported using the height of the bed, such as H = 1.5 cm [29–31].

The existence of a possible adjustment that could describe the adsorptive process was present, specifically in the mass transfer zone. Table 2 shows variables of the TM ($q_{TH}$, $K_{TH}$, $R^2$) and YNM ($\tau_{0.5Co}$, $K_{YN}$, $R^2$) for $C_{-conv}$ and $C_{-FeCl_3}$ (Bordeaux B).

**Table 2.** Variables of the TM ($q_{TH}$, $K_{TH}$, $R^2$) and YNM ($\tau_{0.5Co}$, $K_{YN}$, $R^2$) for $C_{-conv}$ and $C_{-FeCl_3}$. Bordeaux B.

| TM | $q_{TH}$, mg/g | $k_{TH}$, mL/mg·min | $R^2$ |
|---|---|---|---|
| $C_{-conv}$ | 237.88 | 0.0058 | 0.9842 |
| $C_{-FeCl_3}$ | 216.21 | 0.0080 | 0.9799 |
| **YNM** | $\tau_{0.5Co}$, min | $k_{YN}$, min$^{-1}$ | $R^2$ |
| $C_{-conv}$ | 17.97 | 0.2989 | 0.9794 |
| $C_{-FeCl_3}$ | 16.38 | 0.4068 | 0.9799 |

The coefficient of determination values ($R^2$) were very similar in all cases. They showed a good fit to the experimental values for both adsorbents. However, a slightly better TM fit was shown compared to the YNM (TM: $R^2$ > 0.9799). There were no significant differences between both model values. The TM and YNM showed a good fit to the experimental values for both adsorbents. In addition to this, the process followed the Langmuir adsorption–desorption kinetics without axial dispersion (TM). The adsorbate maximum solid-phase concentration on the adsorbent ($q_{TH}$) and rate constant $K_{TH}$ were determined using data obtained from columns continuous studies with the Thomas adsorption model. The Bordeaux maximum adsorbed solid-phase concentration was $C_{-conv}$ (237.88 mg/g) > $C_{-FeCl_3}$ (216.21 mg/g). These values were higher than those published by Lakshmipathy and Sarada [32]. They reported a blue methylene maximum solid-phase concentration of wastewater melon rind of $q_{TH}$ = 135.5 mg/g [33]. Elgarahy et al. [34], reported the maximum sorption reaching 87.69 mg/g for methylene blue on marine algae through batch and continuous processes in solutions [34]. Consequently, the time required for a 50% adsorbate breakthrough or time when Ct/Co = 0.5 ($\tau_{0.5Co}$) was 17.97 min ($C_{-conv}$) and 16.38 min ($C_{-FeCl_3}$).

The Red G adsorptive performance showed that an average sum of errors (S) of all experimental values along the breakthrough curve from $t$ = 0, with $C_{-conv}$ and $C_{-FeCl_3}$,

was $\Sigma \, \overline{X} \, S_{(Cef/Co)}$ = ±0.01 and ±0.02, respectively, which showed little dispersion and reproducibility in the repetitions carried out.

The results of the evaluation parameters of the fixed-bed column packed *with C$_{-conv}$* and C$_{-FeCl_3}$ during the adsorption process of Red G in the influent solution are shown in Table 3.

**Table 3.** Evaluation parameters of the fixed-bed column packed with $C_{-conv}$ and $C_{-FeCl_3}$ during the adsorption process of Red G in the influent solution.

| Adsorbent | $t_b$/min | $t_s$/min | $L_{ZTM}$/cm | $q_{total}$/mg | $q_{eq}$/mg · g$^{-1}$ | $M_{total}$, mg | $R_T$/% |
|---|---|---|---|---|---|---|---|
| $C_{-conv}$ | 11.54 | 238.46 | 8.19 | 11.07 ± 0.13 | 2.21 ± 0.03 | 16.30 ± 0.00 | 67.87 ± 0.78 |
| $C_{-FeCl_3}$ | 10.54 | 250.00 | 8.06 | 11.99 ± 0.90 | 2.44 ± 0.24 | 17.44 ± 0.64 | 68.70 ± 2.63 |

Note: $t_b$, breakthrough time when $C_{ef} \approx 0.03C_0$. $t_s$, saturation time when $C_{ef} \approx 0.82C_0$.

The average values obtained from the breakthrough time ranged between 10.54 and 11.54 min for the removal on C$_{-FeCl_3}$ and C$_{-conv}$, respectively. The tb achieved in the removal of Red G on C$_{-FeCl_3}$ was approximately one minute before that obtained with C$_{-conv}$, which translated into a shorter life time in the treatment for this bed. Regarding the averages of the saturation time, ts, an increase was observed from a ts = 238 min, reached with the C$_{-conv}$ ($V_{treated\,ef}$ = 360 mL), to a ts = 250 min, obtained in the removal of brown on C$_{-FeCl_3}$ ($V_{treated\,ef}$ = 385 mL).

The average lengths of the $L_{ZTM}$ (mass transfer zone) in the influent solution were found between 8.06 and 8.19 cm. The existence of some possible adjustments that could describe the adsorptive process was evidenced, specifically in the mass transfer zone.

Table 4 presents the variables of the TM ($q_{TH}$, $K_{TH}$, $R^2$) and YNM ($\tau_{0.5Co}$, $K_{YN}$, $R^2$) for C$_{-conv}$ and C$_{-FeCl_3}$ Red G.

**Table 4.** Variables of the TM ($q_{TH}$, $K_{TH}$, $R^2$) and YNM ($\tau_{0.5Co}$, $K_{YN}$, $R^2$) for $C_{-conv}$ and $C_{-FeCl_3}$ Red G.

| *TM* | $q_{TH}$, mg/g | $k_{TH}$, mL/mg·min | $R^2$ |
|---|---|---|---|
| $C_{-conv}$ | 338.46 | 0.0037 | 0.9944 |
| $C_{-FeCl_3}$ | 329.42 | 0.0030 | 0.9925 |
| **YNM** | $\tau_{0.5Co,}$ min | $k_{YN}$, min$^{-1}$ | $R^2$ |
| $C_{-conv}$ | 28.74 | 0.1690 | 0.9944 |
| $C_{-FeCl_3}$ | 26.63 | 0.1481 | 0.9975 |
| *TM* | $q_{TH}$, mg/g | $k_{TH}$, mL/mg · min | $R^2$ |
| $C_{-conv}$ | 237.88 | 0.0058 | 0.9842 |
| $C_{-FeCl_3}$ | 216.21 | 0.0080 | 0.9799 |
| **YNM** | $\tau_{0.5Co,}$ min | $k_{YN}$, min$^{-1}$ | $R^2$ |
| $C_{-conv}$ | 17.97 | 0.2989 | 0.9794 |
| $C_{-FeCl_3}$ | 16.38 | 0.4068 | 0.9799 |

The coefficient of determination values ($R^2$) was very similar in all cases, but these values were higher than those obtained with the Red G dye. They showed a good experimental value fit for both adsorbents. However, YNM showed a slightly better fit compared to the TM (YNM: $R^2$ > 0.9944). The TM and YNM showed a good experimental value fit for both adsorbents. According to the TM, similar to Bordeaux B, the process followed the Langmuir adsorption–desorption kinetics without axial dispersion (TM). The maximum solid-phase concentration of Red G adsorbed was C$_{-conv}$ (338.46 mg/g) > C$_{-FeCl_3}$ (329.42 mg/g). These values were higher than those published by Lakshmipathy and Sarada [32]. Similar results were found by Bhattacharjee et al. [33] and Elgarahy et al. [34] during methylene blue removal using different sorbents. Consequently, the time required for a 50% adsorbate breakthrough or time when Ct/Co = 0.5 ($\tau_{0.5Co,}$) was 28.74 min (C$_{-conv}$) and 26.63 min

($C_{-FeCl_3}$). These results were higher than the ones obtained for Bordeaux B. Similar results were found by Usman and Khan [35] when they adsorbed anionic dye from wastewater using a polyethyleneimine-based macroporous sponge in batch and continuous studies.

Figure 8A shows the pH performance during the adsorption process in the fixed-bed column. The mean Σ of the standard deviation of all experimental points for pH $C_{-conv}$ and $C_{-FeCl_3}$ was ±0.27 and ±0.51, respectively (Bordeaux B).

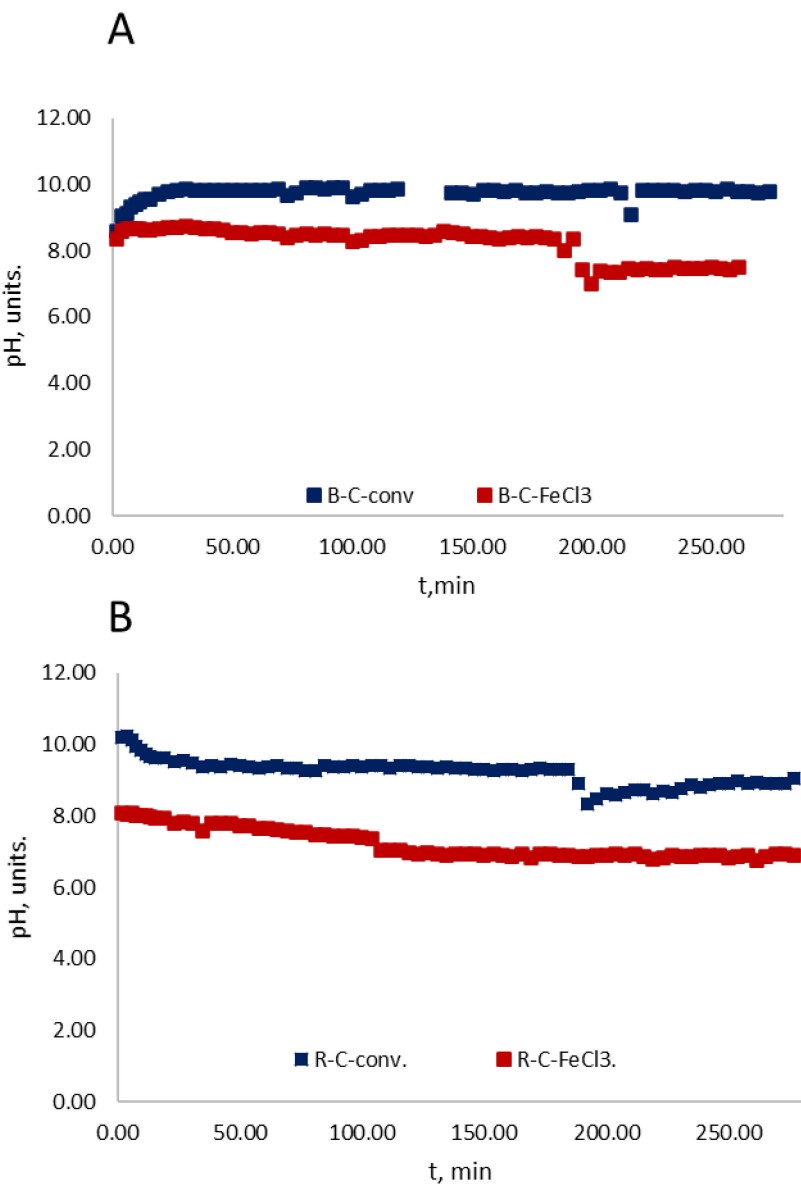

**Figure 8.** (**A**) Bordeaux B and (**B**) Red G pH performance outside the adsorption column with respect to time in minutes on $C_{-conv}$ and $C_{-FeCl_3}$.

The Bordeaux B solution pH values on the $C_{-conv}$ were in a range between 9 and 10 units, which implied that the activation process of this carbon was in basic medium, while the performance of the pH on the modified carbon $C_{-FeCl_3}$ was in a range of 7 and 8 units, both showing a fairly stable trend.

Figure 8B shows the pH performance of Red G during the adsorption process in the fixed-bed column. The mean Σ of the standard deviation of all experimental points for pH $C_{-conv}$ and $C_{-FeCl_3}$ was ±0.27 and ±0.51, respectively. The pH values of the Red G solution on $C_{-conv}$ were in a range between 9 and 10 units, which implied that the activation process

of this carbon was a basic medium, while the performance of the pH on the modified carbon $C_{-FeCl_3}$ was in a range of 7 and 8 units, both showing a fairly stable trend.

### 3.2.3. Economic Study

An economic analysis was performed. The adsorptive system filtration capital cost (CCFS), including total estimated manufacturing, installation and testing costs, was determined based on current purchasing prices. The prices were: installation and testing, 10 USD; frame, 5 USD; valves and hoses, 3 USD, peristaltic pump, 100 USD; total CCFS, 118 USD. The maintenance cost (MC) and operating cost (OC) were assumed to be 20% of the CCFS (47.2 USD). The electric power cost could be determined based on the kWh Peru price in 2021, which was approximately 0.220 USD/kWh.

### 3.2.4. SEM Analysis

Representative $C_{-FeCl_3}$ SEM micrograms before and after Bordeaux B and Red G dye adsorption are depicted in Figure 9. Figure 9A revealed the presence of pronounced and well-organized voids due to the activated process [36]. The surface presented cracks and holes in its morphological structure and small cavities indicating availability of pores for adsorption. These pores may be the area where dye adsorption took place on the adsorbent surface. Figure 9B,C are a product of the continuous dye adsorption on the activated carbon surface, which was part of the fixed bed during the adsorption operation. The formation of a layer or crust of irregular characteristics was observed on the surface of the activated carbon.

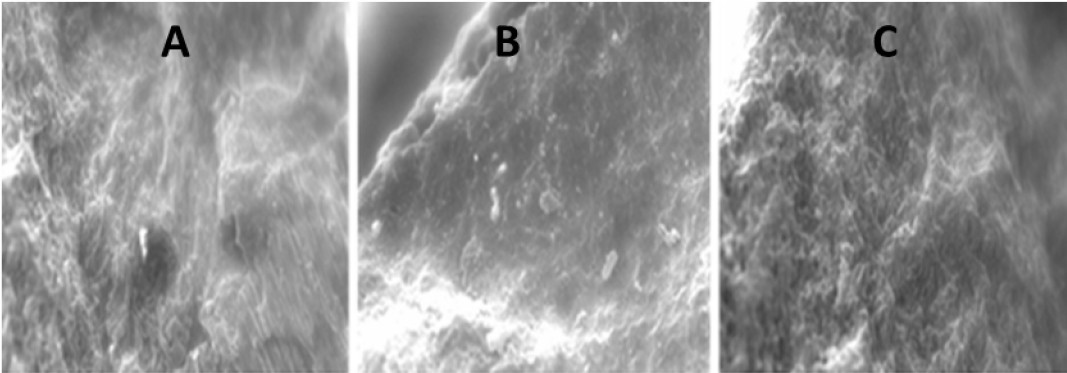

**Figure 9.** SEM images of (**A**) $C_{-FeCl_3}$, (**B**) Bordeaux B and (**C**) Red G dye adsorption.

### 3.3. Other Alternative Processes

As alternative treatment processes, FeOCl and $H_2O_2$ were added as potential catalysts (10 mg/L) to each dye, stirring for 30 min at 200 rpm in the presence of light (Vis LED Photo Reactor), and evaluating the progress at different reaction times [37]. The results (photo-Fenton) showed a progressive degradation of both dyes under study during a reaction time of 150 min. A slight tendency to decrease the degradation was observed after 120 min, obtaining a greater tendency in the degradation of Red G after the first 80 min. After 150 min of reacting, the order of degradation was as follows: Red G: 78.26 ± 6.52 > Bordeaux B: 68.62 ± 7.62)%. The comparison analysis of means in related samples revealed that there were no statistically significant differences among them ($p > 0.05$) when the degradation percentages were compared in a 150 min timeframe (Figure 10A).

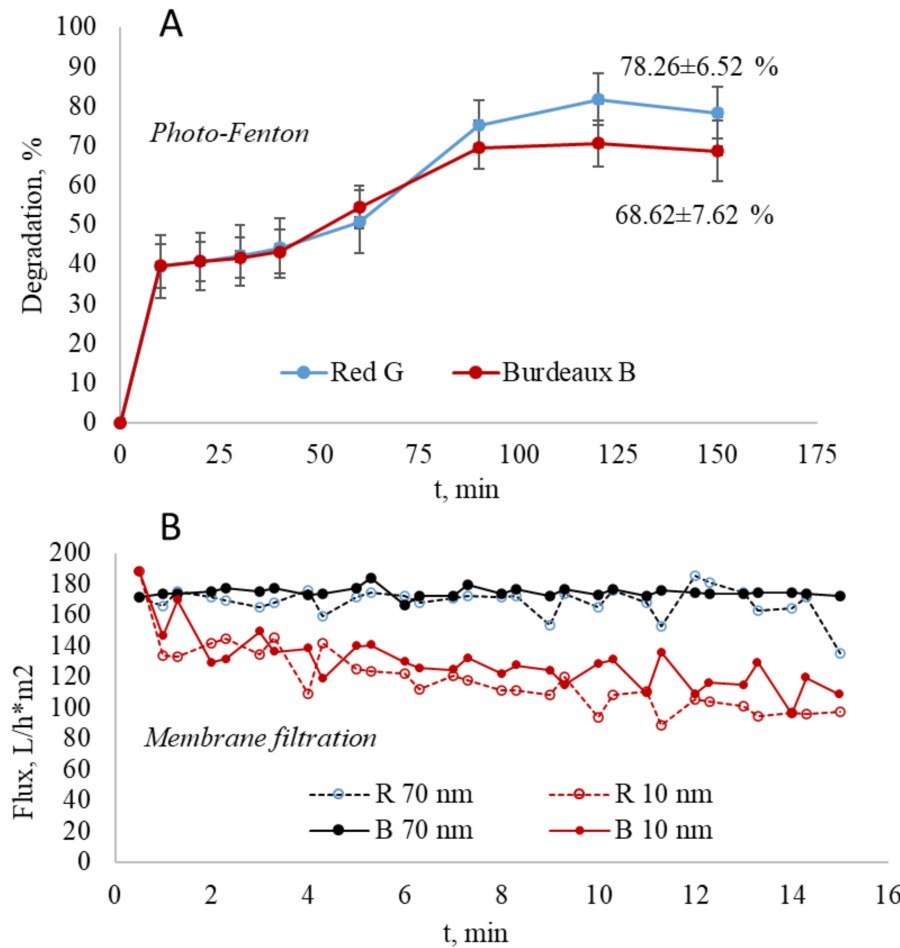

**Figure 10.** (**A**) Bordeaux B and Red G degradation applying photo-Fenton process; (**B**) membrane filtration flux using two pore sizes (70 and 10 nm).

On the other hand, a filtration membrane was applied in order to use as an inorganic tubular membrane. The flow performance is shown in Figure 10B. Zhang et al. [38], obtained a $61.0 \pm 18.3$ L m$^{-2}$ h$^{-1}$ bar$^{-1}$ water flux of the optimal membrane, and the removal efficiency was $91.4 \pm 0.8\%$ when they used membranes based on ultrathin polyethylene porous substrates for the continuous removal of anionic dyes. In this case, the flux average was the highest when dyes were treated. Red G showed lower values than Bordeaux B.

## 4. Conclusions

The removal of Red G and Bordeaux B dyes (alpaca wool dyes) from water using a packed-bed column with $C_{-conv}$ and $C_{-FeCl_3}$ was explored. The dimensioned molecular structure of these dyes was complex, presenting different sizes, shapes and functional groups, influencing the adsorptive process. The Red G adsorption capacity was higher on both adsorbents; $q_{TH}$ = (338.46 and 329.42) mg/g. There were two important physicochemical differences between $C_{-conv}$ and $C_{-FeCl_3}$, including $C_T$ = $(10.53 \pm 0.12$ and $8.98 \pm 0.21)\%$ and pH = $(8.55 \pm 0.05$ and $4.14 \pm 0.03)$ units. The sorption on $C_{-FeCl_3}$ (higher reactive surface) was lower than $C_{-conv}$ for both dyes. Factors such as high pH conditions (favored pH > 8), the large presence of groups in the dyes (C–O, SO$_3$, C=C, -N=N–, SO$_2$ and $Cr^{+3}$, $Na^+$ atoms interacting in a noncovalent way), the structure and the molecular size due to their electron clouds were likely to improve the sorption capacity on less carbon-reactive surfaces as in $C_{-conv}$. In general, the sorption capacity was good, presenting a removal efficiency of over 55%.

**Author Contributions:** G.J.C.A.: Conceptualization, Methodology, Validation, Investigation, Writing—Original Draft, Writing—Review & Editing. J.M.V.Q.: Investigation, Data curation, Conceptualization. R.T.H.: Validation, Writing—Review & Editing. K.T.M.: Validation, Writing—Review & Editing. A.C.M.V.: Validation, Investigation, Data curation, Conceptualization. J.A.A.-P.: Validation, Investigation, Data curation, Conceptualization. J.D.C.G.: Validation, Writing—Review & Editing. D.A.P.T.: Validation, Writing—Review & Editing. All authors have read and agreed to the published version of the manuscript.

**Funding:** This study was supported by the Consejo Nacional de Ciencia, Tecnología e Innovación Tecnológica (CONCYTEC) and the Programa Nacional de Investigación Científica y Estudios Avanzados (PROCIENCIA)—Perú. Grant N° 06-2019-FONDECYT-BM-INC.INV.

**Conflicts of Interest:** All the authors declare no conflict of interests.

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
