# Peer review of "Enhanced Removal of Bordeaux B and Red G Dyes Used in Alpaca Wool Dying from Water Using Iron-Modified Activated Carbon"

_water, doi:10.3390/w14152321_

Round 1

Reviewer 1 Report

Abstract:

Line 15: on conventional carbon

Introduction:

Line 26;  It accounts for

Line 38: Activated carbon has been shown to be

Results and Discussion

Line 347 : Iheanacho et al., 2021

Line 349  Breakthrough curves of the adsorption of the dyes  on fixed beds of C-conv and ?−????3. This is an incomplete sentence.

Line 392 There were no significant differences between both models values.

Line 399 These values are higher than those published by Lakshmipathy and Sarada (2016).

Line 401- 403   Elgarahy et al., (2021), reported 401 the maximum sorption, reaching 87.69 mg/g for the methylene blue on marine algae by  batch and continuous processes in solutions.

Line 439 concentration of Red G

Line 440 These values are higher than those published by Lakshmipathy and Sarada (2016).

Line 455 performance of the Reg G,

Line 492 when they used membranes

Conclusion

Line 500 Granular activated carbon (GAC) was found

Line 507 Results suggested  that FeCl3 modified GAC

Reviewer 2 Report

The paper „Enhanced removal of Burdeaux B and Red G dyes used in alpaca wool dying from water using iron-modified activated carbon” is interesting, but needs many corrections.

I propose to edit the abstract. Please provide the most important information (purpose, scope of research and the most important results and conclusions). The first sentence is unnecessary (it fits better with the introduction). Half of the abstract are the parameters of sorption models (most of them are redundant - e.g. the coefficient of determination). Please focus on the most important parameters, e.g. sorption capacity.

The abstract also uses the abbreviation "GAC" - it is only explained in the "Conclusions". I suggest using the abbreviation ?????3 (which was previously given in the abstract) instead.

There is also information in the abstract that ?????3 is a potential sorption material for dyes (but the results show that "C-conv" is not worse than ?????3). The Thomas model even shows that "C-conv" is better than "?????3" because it has a greater sorption capacity.

Overall, in section 3 (Results and Discussion), I suggest you comment more on the differences between "C-conv" and "?????3". Please make clear whether the modification has proved beneficial.

Other comments:

Page 1, Line 26

Mr. Cheng in his article mentions 700,000 tons per year (not 10,000 tons per year). However, these data are most likely out of date. Currently, it is rather 1,000,000 tons per year.

Page 1, Line 31

Instead of "adsoption" it should rather be "adsorption".

Page 2, Methods, 2.1 Reagents

Please provide here the parameters of the tested dyes: chemical formula, chemical nature (anionic / cationic), dye class (e.g. is it an azo dye), molar mass, characteristic functional groups + drawing with structural formulas.

Please also provide the parameters of the activated carbon (e.g. porosity, specific surface area).

Page 6, Line 207

Presumably, instead of "Figure 5" it should be "Figure 4".

Page 10, Line 317

What is "LZTM"? - please explain.

Page 11, Line 354

Most of these parameters have already been mentioned in Section 2.2 (Mtehods). I think that repeating the same information in the results is unnecessary.

Page 11, Figure 7 B

There are no points (experimental data) for Red G. in the graph.

Page 12, Line 371

Why (on what basis) is it assumed that the saturation time is when Cef = 0.82 C0?

Page 12, Line 390

"R2" - This is the coefficient of determination (not correlation). The correlation coefficient is "R" (R only).

 Page 14, Line 440

It would be interesting if the authors compared their own results with those of other authors in the table.

Page 15, Figure 8

What's the point of having 2 of the same axes (same scale and unit) on both sides of the chart?

Page 16, punkt 3.4

This point seems redundant. I think that the article on dye sorption should not include the results of other decolorization processes. Instead, I advise you to expand the discussion of the results on dye sorption (compare your own results with those of other authors).

Page 17, Conclusion

Why are authors only now using the abbreviation "GAC"? Why are "C-conv" and "?????3" not used? As the article is modified activated carbon, the conclusions should include a conclusion about the influence of the modification on the sorption capacity of activated carbon.

Reviewer 3 Report

In this work, Iron-modified activated carbon was explored to remove Red G and Burdeaux B dyes from water using packed bed column. The work is suitable to be published in Water. Will you please pay close attention to the comments below, and provide a detailed response to each one.

1.      The measurement of molecular size in the manuscript is not strict. Does the author consider essential factors such as van der Waals radius?

2.      The specific surface area of the material is an essential factor affecting the adsorption effect. It is suggested to add the BET test.

3.      What was the pH of the solution at equilibrium?

4.      Authors should provide the economic study of the production for commercial viability.

5.      The author should carefully prepare the Figures for the article. The current Figures are not good enough in terms of the resolution and standardization.

6.      Is the stable material in an acidic environment? The author should give relevant conclusions.

Round 2

Reviewer 2 Report

The manuscript has been revised. I think the article can be accepted.

Author Response

Thank you for your kindness and your evaluation. 

Reviewer 3 Report

After the first run of revisions, the manuscript improved in quality. However, I will recommend it to be published until the following comments are addressed.

1.     Compared with the previous work, the author should clearly explain the improvement of this work.

2.     Is the stable material in an acidic environment? The author should give relevant experience. Otherwise, the material is difficult to be applied in practical.

3.     N2 adsorption/desorption isotherms should be added.
